# The Antifungal Effects of Citral on *Magnaporthe oryzae* Occur via Modulation of Chitin Content as Revealed by RNA-Seq Analysis

**DOI:** 10.3390/jof7121023

**Published:** 2021-11-29

**Authors:** Xingchen Song, Qijun Zhao, Aiai Zhou, Xiaodong Wen, Ming Li, Rongyu Li, Xun Liao, Tengzhi Xu

**Affiliations:** 1Institute of Crop Protection, Guizhou University, Guiyang 550025, China; songxingc@yeah.net (X.S.); qijunzhao@126.com (Q.Z.); aachau326@126.com (A.Z.); 15117713970@163.com (X.W.); lm21959@163.com (M.L.); liaoxun2012@163.com (X.L.); xtz9012@126.com (T.X.); 2College of Agriculture, Guizhou University, Guiyang 550025, China; 3The Provincial Key Laboratory for Agricultural Pest Management in Mountainous Region, Guiyang 550025, China

**Keywords:** *Magnaporthe oryzae*, citral, RNA-seq, antifungal mechanism, gene expression

## Abstract

The natural product citral has previously been demonstrated to possess antifungal activity against *Magnaporthe oryzae*. The purpose of this study was to screen and annotate genes that were differentially expressed (DEGs) in *M. oryzae* after treatment with citral using RNA sequencing (RNA-seq). Thereafter, samples were reprepared for quantitative real-time PCR (RT-qPCR) analysis verification of RNA-seq data. The results showed that 649 DEGs in *M. oryzae* were significantly affected after treatment with citral (100 μg/mL) for 24 h. Kyoto Encyclopedia of Genes and Genomes (KEGG) and a gene ontology (GO) analysis showed that DEGs were mainly enriched in amino sugar and nucleotide sugar metabolic pathways, including the chitin synthesis pathway and UDP sugar synthesis pathway. The results of the RT-qPCR analysis also showed that the chitin present in *M. oryzae* might be degraded to chitosan, chitobiose, N-acetyl-D-glucosamine, and β-D-fructose-6-phosphate following treatment with citral. Chitin degradation was indicated by damaged cell-wall integrity. Moreover, the UDP glucose synthesis pathway was involved in glycolysis and gluconeogenesis, providing precursors for the synthesis of polysaccharides. Galactose-1-phosphate uridylyltransferase, which is involved in the regulation of UDP-α-D-galactose and α-D-galactose-1-phosphate, was downregulated. This would result in the inhibition of UDP glucose (UDP-Glc) synthesis, a reduction in cell-wall glucan content, and the destruction of cell-wall integrity.

## 1. Introduction

The fungus *M. oryzae* causes rice blast, which is the most widespread and serious disease affecting rice [1,2] and one of the most destructive and economically harmful diseases affecting rice production worldwide [3,4]. At present, the methods of disease-resistant variety breeding and chemical control are mainly used for the prevention and control of *M. oryzae* in China [5]. As a result of the shortcomings of traditional breeding of disease-resistant varieties, such as long breeding cycles, easy-to-lose resistance, and the high cost of research and development in the field of chemical pesticides, pesticide residues, and drug resistance [6], a move away from the traditional excessive dependence on chemical pesticides during crop management became necessary. Under the premise of stabilizing production, intensive efforts were needed to develop biological pesticide technology with low toxicity and low residue characteristics.

The discovery of antifungal and germicidal compounds from plants has led to the efficient creation of new pollution-free pesticides [7,8]. Many plants have antifungal activities, which are related to their antifungal constituents, including alkaloids, terpenes, polysaccharides, esters, ketones, and quinones [9,10,11], including lemongrass, cilantro, cassia, and cinnamon [12]. In particular, citral is an open-chain monoterpene extracted from *Litsea cubeba* oil, which has good antifungal, insecticidal, and antioxidant activities [13]. 

Our previous studies demonstrated that citral has significant biological activity against *M. oryzae* in vitro and in vivo, suggesting its potential application in controlling rice blast [14,15]. Citral may act by disrupting the cell-wall integrity and membrane permeability of *M. oryzae*, thus resulting in physiological changes and cytotoxicity [16]. Importantly, the inhibitory effect of citral on *M. oryzae* appears to be associated with its effects on reducing sugar, soluble protein, pyruvate, and malondialdehyde contents, and chitinase activity, in mycelia [15,17]. However, the gene targeted by citral and thus responsible for its activity in *M. oryzae* has not been identified. Recent advances in high-throughput genomic technology, including transcriptomics, have provided a new method for studying fungal growth, pathogenesis, and the genes involved in resistance [18,19,20]. Thus, we decided to study the gene expression profile of citral-treated *M. oryzae* to gain some insight into its mode of action. In this research, the objectives were (i) to screen and annotate the differentially expressed genes (DEGs) of *M. oryzae* after treatment with citral using RNA-seq, (ii) to ascertain the expression profiles of these *M. oryzae* DEGs, and (iii) to use the information about DEGs and their expression profiles to construct a model diagram illustrating the mechanism of citral acting on *M. oryzae*. Furthermore, the samples were reprepared for real-time fluorescence quantitative verification of the RNA-seq data. The inhibitory targets of citral in *M. oryzae* were obtained through the changes in the DEGs levels and RT-qPCR analysis verification results. As far as we know, this is the first report that uses the RNA-seq method to study the mechanism of citral in *M. oryzae*. Studies of the closely related response genes and signaling pathways can help elucidate the molecular mechanism of citral acting on *M. oryzae*, and provide a scientific basis for its mode of action and a valuable reference for the control of *M. oryzae*.

## 2. Materials and Methods

### 2.1. Strain and Growth Conditions

*M. oryzae* strain ZB15 was isolated from diseased rice panicles in Guizhou Province, China, and stored on potato dextrose agar (PDA) at 4 °C for 7 days. A 5 mm diameter agar disc was taken from the colony margin using a punch, inoculated on a cellophane sheet-covered PDA plate, and cultured at 28 °C for 15 days. The hyphae were then made into a suspension containing citral at concentrations of 0, 50, and 100 µg/mL. The inoculum was cultured at 28 °C with stirring at 180 rpm. The mycelia were collected by filtering through two layers of cotton gauze and washed with distilled deionized water at 0, 9, 12, and 24 h post incubation [21].

### 2.2. RNA Extraction and Preparation of cDNA Library

Total RNA was isolated using TRIzol reagent (Invitrogen, Carlsbad, CA, USA) following the manufacturer’s instructions. The yield and purity of each RNA sample were quantified using a NanoDrop ND-1000 (NanoDrop, Wilmington, DE, USA). The RNA integrity was assessed as a function of the RNA integrity number (RIN) >7.0, using the Agilent 2100 bioanalyzer.

Poly (A) RNA was purified from total RNA (5 µg) using poly-T oligo-attached magnetic beads using two rounds of purification. Then, the poly (A) RNA was fragmented into small pieces using divalent cations under high temperature. Thereafter, the cleaved RNA fragments were reverse-transcribed to create the cDNA, which were then used to synthesize U-labeled second-stranded DNAs with *E. coli* DNA polymerase I. RNase H and dUTP A-bases were added to the blunt ends of each strand, preparing them for T-base overhang adapter ligation. Single or dual index adapters were ligated to the fragments, and size selection was performed with AMPureXP beads. After heat-labile UDG enzyme treatment of the U-labeled second-stranded DNAs, the ligated products were amplified using PCR under the following conditions: initial denaturation at 95 °C for 3 min, followed by 8 cycles of denaturation at 98 °C for 15 s, annealing at 60 °C for 15 s, and extension at 72 °C for 30 s; with a final extension at 72 °C for 5 min. The average insert size for the final cDNA library was 300 bp (±50 bp). Finally, the 150 bp paired-end sequencing was performed on an Illumina Hiseq 4000 (LC Bio, Hangzhou, China). The bioinformatics analysis pipeline is shown in Appendix A.

### 2.3. Sequence Assembly and Annotation

The raw data generated by sequencing needed to be preprocessed. Cutadapt was used to filter out unqualified sequences and obtain valid data before further analysis [22]. Reads with an adaptor containing more than 5% of N (N indicates that base information cannot be determined) were removed. Low quality reads (more than 20% of the total read exhibiting a mass value of Q ≤ 10) were removed. The original sequencing amount, the effective sequencing amount, Q20, Q30, and GC contents were counted and comprehensively evaluated. Thereafter, sequence quality was verified using FastQC (http://www.bioinformatics.babraham.ac.uk/projects/fastqc/, accessed on 21 November 2021). The known annotation information in the species database (https://www.ncbi.nlm.nih.gov/genome/?term=Magnaporthe%20oryzae, accessed on 21 November 2021) was analyzed, including chromosome number, gene number, transcript number, GO annotation (http://geneontology.org, accessed on 21 November 2021), and KEGG annotation (http://www.kegg.jp/kegg, accessed on 21 November 2021).

### 2.4. Gene Expression Analysis

The algorithm feature Counts version 1.5.0-p3 was used for the FPKM calculation of each gene, according to the length of the gene and the read count mapped to this gene [23]. Expression levels of different samples of *M. oryzae* treated with citral were obtained by calculating a whole-genome model [24,25]. We used edgeR to finish the StringTie assembly and quantitative genetic variance analysis (significant difference threshold for |log2 foldchange| ≥ 1, *p* < 0.05) [26]. Finally, the volcano figure, GO, and KEGG enrichment analyses were performed on the DEGs.

### 2.5. Quantitative Real-Time PCR Analysis

The total RNA for RT-qPCR analysis was extracted using a 2× SG Fast qPCR Master Mix (Sangon, Shanghai, China), and 1.0 μg of RNA was used for reverse transcription with an M-MuLV First-Strand cDNA Synthesis Kit (Sangon, Shanghai, China) in a 20 μL reaction volume according to the manufacturer’s instructions. Gene-specific primer pairs were designed using the Sangon primer design and synthesis online platform (https://www.sangon.com/newPrimerDesign, accessed on 21 November 2021) on the basis of the transcript sequence and synthesized by Sangon Biotech Co., Ltd. (Shanghai, China) (Table 1). The actin gene was used as an internal control [27]. The PCR was performed using SYBR Green’s method and a Bio-Rad CFX96 real-time PCR system (Bio-Rad, CA, USA) with a total of 96 well plates, each sample was set up with three biological replicates. The final volume for each reaction was 20 μL with the following components: 2 μL cDNA template (1 ng/μL), 10 μL 2× SG Fast qPCR Master Mix (Sangon, Shanghai, China), 0.4 μL forward primer (200 nM), 0.4 μL reverse primer (200 nM), and 7.2 μL PCR-grade water. The reaction was conducted under the following conditions: 95 °C for 3 min, followed by 40 cycles of denaturation at 95 °C for 10 s and annealing/extension at 56 °C for 30 s. The melting curve was obtained by heating the amplicon from 65 °C to 95 °C at increments of 0.5 °C per 5 s. The relative quantification of gene expression was computed using the 2^−ΔΔCt^ method.

### 2.6. Gene Expression of the Genes Related to Amino Sugar and Nucleotide Sugar Metabolism

In order to verify the effect of amino sugar- and monosaccharide metabolism-associated gene expression on the cell wall, eight DEGs from the amino sugar and monosaccharide metabolism pathways were selected (Appendix A), and their effects on the cell wall were verified according to the method described in Section 2.5. The differential expression of *M. oryzae* genes was determined by RT-qPCR after treatment with citral (100 µg/mL) for 24 h.

## 3. Results

### 3.1. Read Generation and De Novo Assembly

Thirty-six cDNA libraries prepared from *M. oryzae* were sequenced using the Illumina Novaseq™ 6000 platform. A total of 16.08 million raw reads were generated. After removing adapters and filtering the low-quality sequences, a total of 1,330,379,916 clean reads were generated from the libraries of *M. oryzae* treated with citral of 0 µg/mL (a0, A2, A3, A4), 50 µg/mL (b0, B2, B3, B4), and 100 µg/mL (c0, C2, C3, C4). The sequencing depth of each sample was 4.68–6.69 Gb. The proportion of clean reads ranged from 75.23% to 88.76% (Appendix A). More than 91.54% clean reads could be mapped to the reference genome. Among the mapped reads, more than 85.33% were uniquely mapped, and more than 97.81% were mapped to gene exon regions (Appendix A). The results showed that the sequencing quality of mapped reads was higher, the assembly integrity was better, and the credibility was higher than for clean reads. Therefore, they could be used for subsequent quantitative expression and annotation analysis.

### 3.2. Differentially Expressed Gene Analysis

The DEGs from the 36 transcriptome libraries were analyzed. We used log10 (FPKM + 1) for gene expression display. The maximum gene expression FPKM value was 36,563.15 (Appendix A). The global analysis of *M. oryzae* genes that were differentially expressed in response to treatment with citral was conducted using criteria of |log_2_ ratio| ≥ 1 and *p* < 0.05, and the results were visualized using a volcano plot and Venn diagram. There were 25 and 73 DEGs that overlapped across the four time points in *M. oryzae* after treatment with 50 and 100 μg/mL citral, respectively (Figure 1A,B). As shown in Figure 1C, the number of DEGs increased after treatment with 100 μg/mL citral compared with the 50 μg/mL citral treatment for 24 h. We identified 1224 upregulated genes and 3431 downregulated genes after treatment with 50 μg/mL citral for 24 h and 1009 upregulated genes and 2084 downregulated genes after treatment with 100 μg/mL citral for 24 h (Figure 2 and Figure 3). The results show that the numbers of common DEGs (2301) for all concentrations and time points of citral treatment and specific DEGs (649) in *M. oryzae* after treatment with 100 μg/mL citral for 24 h were the largest among all the treatment groups.

### 3.3. Functional Annotation and Classification

The possible functions of the identified DEGs were predicted using the gene function classification system GO, which has three ontologies, namely, molecular functions, cellular components, and biological processes. We used GO analysis to classify the functions of genes in *M. oryzae*. A total of 10,537 annotated genes were categorized into 50 GO functional groups (Figure 4C). Most of the genes were enriched in the molecular function category, followed by the biological process and cellular component categories, as shown in Figure 4C. Within the biological process category, “mycelium development”, and “conidium formation” featured the highest number of genes. On the other hand, “cellular component”, “nucleus”, and “integral component of membrane” were the most enriched terms in the cellular component category. The “zinc ion binding” term was the most prevalent in the molecular function category (Figure 4A). For KEGG analysis, a total of 4073 annotated genes were used for mapping in *M. oryzae*; the main active pathways involved were: ribosome biogenesis in eukaryotes, glyoxylate and dicarboxylate metabolism, lysine degradation, and glycolysis. The largest category was purine metabolism (181, 4.4%), followed by biosynthesis of amino acids (175, 4.2%), meiosis in yeast (169, 4.1%), cell cycle in yeast (163, 4.0%), and carbon metabolism (159, 3.9%). These annotations provide a further understanding of the transcriptome data, as well as their functions and pathways, in *M. oryzae*.

### 3.4. RNA-Seq Validation by RT-qPCR

To confirm the accuracy of the sequencing results, we selected eight genes and used RT-qPCR to determine their relative expression levels in *M. oryzae* under the corresponding treatments. Eight pairs of primers were designed, and *actin* was used as an internal control for normalization (Table 1). The RT-qPCR results indicated that all eight genes were regulated in a manner similar to that revealed from the RNA-Seq data (Figure 5). The RT-qPCR results indicated that the RNA-Seq data were reliable and could be used for subsequent analysis.

### 3.5. Identification of Genes in Amino Sugar and Nucleotide Sugar Metabolism

Functional annotation results for DEGs related to amino sugar and nucleotide sugar metabolism were identified. The pathways of amino sugar and nucleotide sugar metabolism were assigned using KEGG analysis, and they were found to influence the differential expression of enzyme nodes in the chitin formation and UDP glucose synthesis pathways. Nine related enzyme nodes were significantly differentially expressed in aminosaccharide and nucleotide sugar metabolism pathways according to the annotation results. In particular, five enzyme nodes were discovered for the chitin formation pathway: EC:3.2.1.14 (enzyme or protein chitosan (chitobiose)), EC:3.2.1.52 (N-acetyl glucosamine (GlcNAc)), EC:3.5.99.6 (glucosamine 6-phosphate isomerase (GlcN-6P)), EC:2.4.1.16 (chitin synthase), and EC:1.6.2.2 (ferrous cytochrome B5). On the other hand, EC:5.4.2.2 (phosphoglucomutase) and EC:2.7.7.12 (galactose-1-phosphate uridylyltransferase) were discovered for the UDP glucose synthesis pathway (Appendix A). The other two related enzyme nodes were EC:3.2.1.55 (α-N-arabinofuranosidase) and EC:3.2.1.37 (β-xylosidase). However, the eight DEGs were related to the chitin formation and UDP glucose synthesis pathways according to the enzyme node enrichment analysis. The results show that enzyme nodes in the pathways of amino sugar and nucleotide sugar metabolism were differentially expressed in *M. oryzae* after treatment with citral, which was related to chitin formation, along with a certain effect on the cell wall. Thus, the results suggest that the mechanism of citral action against *M. oryzae* may involve damage to the cell wall.

### 3.6. Effect of Citral on Amino Sugar and Nucleotide Sugar Biosynthesis in M. oryzae

In order to further verify the regulatory effect of citral on the chitin synthesis pathway in *M. oryzae*, we selected eight DEGs (six upregulated genes and two downregulated genes) identified in the amino sugar and nucleotide sugar metabolism pathways of *M. oryzae* after treatment with citral. The changes in trends were observed to be similar between RNA-seq and qRT-PCR assays, although several genes presented a difference in expression at the same time point (Figure 6). There were three DEGs upregulated upstream of the chitin synthesis pathway compared with the control group, namely, *MGG_*10333 (class III chitinase), *MGG_*01336 (*Bacteriodes thetaiotaomicron* symbiotic chitinase), and *MGG_*00625 (glucosamine-6-phosphate isomerase), with gene expression upregulated by 17.52-, 3.75-, and 0.07-fold, respectively. The expression levels of *MGG_*01605 and *MGG_*03920, downstream of the chitin synthesis pathway, were also upregulated 0.41- and 2.69-fold, respectively, whereas the expression of *MGG_*09551 (ferrocytochrome B5) downstream of the chitin synthesis pathway was downregulated by 0.65-fold. The expression of *MGG_*04495 was upregulated in the UDP glucose synthesis pathway, involved in the regulation of α-d-glucose 6-phosphate synthesis. Moreover, the expression of *MGG_*05098 was downregulated, involved in the regulation of UDP-α-d-galactose and α-d-galactose-1-phosphate synthesis. The results show that the antifungal mechanism of citral against *M. oryzae* occurs via regulating the expression of key genes involved in the chitin and UDP glucose synthesis pathways.

## 4. Discussion

*M. oryzae* causes huge economic losses around the world. Although the wide-spread use of traditional fungicides can reduce crop loss, it may also potentially harm the environment and human health [28,29,30,31]. Therefore, it is very important to develop safe and environmentally friendly natural pesticide products. At present, microorganisms, microbial products, and biological fertilizers are being developed as biological control agents [32].

Transcriptomics technology allows the changes in RNA levels in micro-organisms to be analyzed under different environments at a molecular level from a macro perspective [33,34,35]. For example, on the basis of the results of RNA-seq, the effects of sodium pheophorbide, a (SPA) treatment, on the expression of three cell-wall-degrading enzyme-related genes in *Pestalotiopsis neglecta* hypha were detected using qRT-PCR, thereby revealing the molecular mechanism underlying the action of SPA on *P. neglecta*. SPA inhibited TCA-related enzyme activity and the expression of three cell-wall-degrading enzyme-related genes [36]. The genome and transcriptome of the pine saprophyte (*Ophiostoma piceae*) were compared with the genome and transcriptomes of the mountain pine beetle-associated pathogen *Grosmannia clavigera*. The differences revealed a major class of pine tree defense compound [37]. Camptothecin (CPT) is a botanical compound primarily isolated from *Camptotheca acuminata* Decne, which is an important medicinal plant in China. CPT is a promising lead compound for developing fungicides against rice blast, as it may bind to the DNA topoisomerase I complex of *M. oryzae*, thus affecting translation and carbohydrate metabolism/energy metabolism, leading to cell death [38]. Our group previously studied citral extracted from *Litsea cubeba* fruit, which had properties consistent with those of leaf oil extracted from Eucalyptus and showed a good inhibitory effect on *M. oryzae* growth [39]. In addition, further research reported the significant antibacterial activity of citral in Aspergillus *ochraceus*, *Penicillium*, *Penicillium roqueforti*, and so on [40,41,42]. In this study, the transcriptome analysis method based on RNA-seq was used to analyze the RNA levels of genes in *M. oryzae* treated with citral. The results show that citral might act to disrupt cell-wall integrity and the cytoplasmic membrane by causing changes in membrane permeability, leading to a loss of intracellular substances and the introduction of wrinkles and depressions on the cell surface.

Chitin is a β-1,4-linked polymer of N-acetylglucosamine and a major component of the fungal cell wall [43,44]. Fungi can successfully invade the host by destroying the conserved chitin in the fungal cell wall. Chitin hydrolase CcCti1 inhibited conidial germination and appressoria formation in *M. oryzae*, thereby affecting its normal growth [45]. In this study, most DEGs were enriched in biological processes related to mycelium development and conidia formation. The polar germination of conidia and the growth of infected hyphae of *M. oryzae* were related to dynamic changes in the *M. oryzae* cell wall. It was found that the expressions of *MGG*_10333 (class III chitinase), *MGG*_01336 (bacteriodes thetaiotaomicron symbiotic chitinase), and *MGG*_00625 (glucosamine-6-phosphate isomerase) were upregulated in *M. oryzae* after treatment with citral. Thus, the chitin present in *M. oryzae* might be degraded to chitosan, chitobiose, N-acetyl-D-glucosamine, and β-D-fructose-6-phosphate following treatment with citral. The expression of FAD-binding domain-containing protein *MGG*_01605 was upregulated in *M. oryzae*, and this gene was found downstream of the chitin synthesis pathway and was involved in the regulation of chitin. The expression of inositolphosphorylceramide-B C-26 hydroxylase *MGG*_03920 was upregulated, involved in ferrocytochrome B5 synthesis. Furthermore, the expression of chitin synthase *MGG*_09551 was downregulated, which inhibited chitin synthesis. Chitin degradation is indicated by damaged cell-wall integrity, as it is an important component of the cell wall in *M. oryzae* [46]. The KEGG pathway analysis showed that citral could affect chitin synthesis and UDP glucose synthesis in the amino sugar and nucleotide metabolic pathways of *M. oryzae* (Figure 7). UDP glucose is a donor in most polysaccharide synthesis pathways and the precursor of glucan synthesis [47]. The UDP glucose synthesis pathway is involved in glycolysis and gluconeogenesis, providing precursors for the synthesis of polysaccharides. The expression of phosphoglucomutase *MGG*_04495 was upregulated in *M. oryzae* after treatment with citral, which was involved in catalyzing the mutual transformation of glucose-6-phosphate and glucose-1- phosphate in the UDP glucose synthesis pathway. Galactose-1-phosphate uridylyl transferase *MGG*_05098, involved in the regulation of UDP-α-D-galactose and α-D-galactose-1-phosphate, was downregulated, which would result in the inhibition of UDP glucose (UDP-Glc) synthesis, a reduction in cell-wall glucan content, and the destruction of cell-wall integrity.

In conclusion, the lipophilic characteristic of citral facilitates its passage and access such that it can disrupt cell-wall integrity and the cytoplasmic membrane by causing changes in membrane permeability, leading to the loss of intracellular substances and the introduction of wrinkles and depressions on the cell surface [48]. The model diagram shows that the gene expression of chitin synthase was downregulated in the chitin formation pathway of *M. oryzae* after treatment with citral, leading to an inhibition of chitin synthesis. On the other hand, the gene expression of chitinase was upregulated, resulting in a reduction in chitin content in the cell wall of *M. oryzae*, leading to damage. However, the expressions of the glucosamine-6-phosphate deaminase and UDP glucose pyrophosphorylase genes were downregulated, resulting in an inhibition of UDP glucose synthesis and a reduction in the cell-wall glucan content, thereby disrupting the *M. oryzae* cell-wall integrity and inhibiting mycelial growth. This study identified the key genes related to cell-wall integrity in *M. oryzae,* which exhibited a response to citral. This further revealed that citral’s excellent antifungal activity against *M. oryzae* occurs by affecting the chitin content, thus providing a scientific basis for its mode of action. The regulatory molecular mechanisms of the transcription factors that mediate chitinase in *M. oryzae* following treatment with citral still require further study.

## 5. Conclusions

An RNA-seq analysis identified the effects of citral concentration on *M. oryzae*. The number of DEGs in *M. oryzae* reached the maximum value after treatment with citral (100 μg/mL) for 24 h. Most DEGs were enriched in biological processes related to mycelium development and conidia formation. KEGG pathway analysis showed that genes in the amino sugar and nucleotide sugar metabolism pathways of *M. oryzae* were differentially expressed following treatment with citral, and these pathways are related to chitin formation. In the chitin synthesis pathway of *M. oryzae*, chitin synthase gene expression was downregulated, resulting in an inhibition of chitin synthesis. Moreover, upregulation of chitinase gene expression can lead to an increase in chitin degradation, thus resulting in a decrease in chitin content and destruction of the cell-wall structure in *M. oryzae*. In the UDP glucose synthesis pathway of *M. oryzae*, UDP glucose synthesis was inhibited, which could reduce the cell-wall glucan content and disrupt the cell-wall integrity. This study further revealed that citral had excellent antifungal activity against *M. oryzae* by affecting the content of chitin, thus providing a scientific basis for elucidating its mode of action. At present, our research group is constructing silencing vectors for the target genes that were screened in this study. Moreover, we are observing the genotype expression and other related mechanisms in the later stages of this process.

## Figures and Tables

**Figure 1 jof-07-01023-f001:**
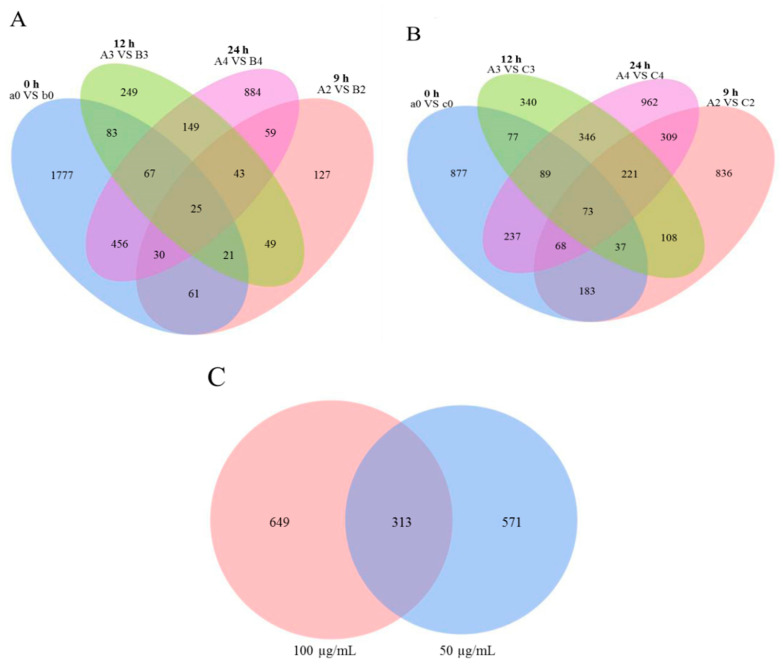
Venn diagram of DEGs in the evaluated libraries. (**A**) Venn map of all DEGs in *M. oryzae* after treatment with citral (50 μg/mL) for the indicated times (h). (**B**) Venn map of all DEGs in *M. oryzae* after treatment with citral (100 μg/mL) for the indicated times (h). (**C**) Venn map of specific DEGs in *M. oryzae* after treatment with citral for 24 h.

**Figure 2 jof-07-01023-f002:**
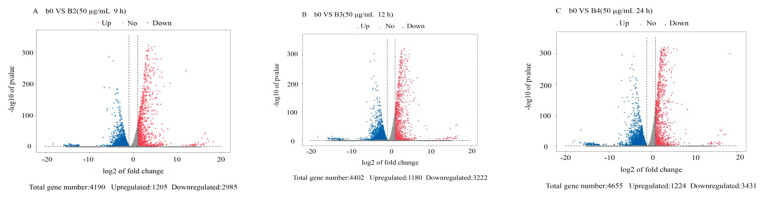
Volcanic map of DEGs in *M. oryzae* after treatment with citral at different concentrations. (**A**) Volcanic map analysis of DEGs in *M. oryzae* after treatment with citral (50 μg/mL) for 9 h. (**B**) Volcanic map analysis of DEGs in *M. oryzae* after treatment with citral (50 μg/mL) for 12 h. (**C**) Volcanic map analysis of DEGs in *M. oryzae* after treatment with citral (50 μg/mL) for 24 h. (**D**) Volcanic map analysis of DEGs in *M. oryzae* after treatment with citral (100 μg/mL) for 9 h. (**E**) Volcanic map analysis of DEGs in *M. oryzae* after treatment with citral (100 μg/mL) for 12 h. (**F**) Volcanic map analysis of DEGs in *M. oryzae* after treatment with citral (100 μg/mL) for 24 h. Red represents upregulated significantly differentially expressed genes, blue represents downregulated significantly differentially expressed genes, and gray dots represent non significantly differentially expressed genes.

**Figure 3 jof-07-01023-f003:**
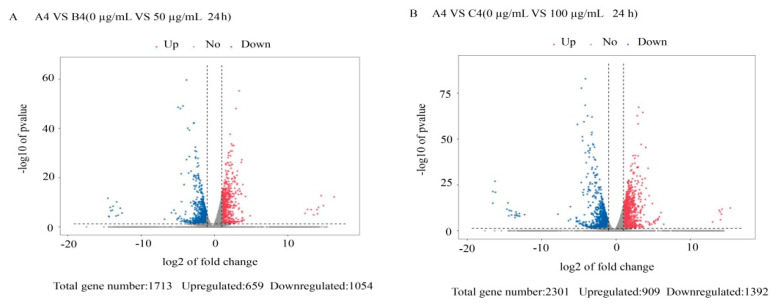
Volcanic map analysis of DEGs in *M. oryzae* after treatment with citral for 24 h. (**A**) Volcanic map analysis of DEGs in *M. oryzae* after treatment with citral (0 μg/mL and 50 μg/mL) for 24 h. (**B**) Volcanic map analysis of DEGs in *M. oryzae* after treatment with citral (0 μg/mL and 100 μg/mL) for 24 h.Red represents upregulated significantly differentially expressed genes, blue represents downregulated significantly differentially expressed genes, and gray dots represent non significantly differentially expressed genes.

**Figure 4 jof-07-01023-f004:**
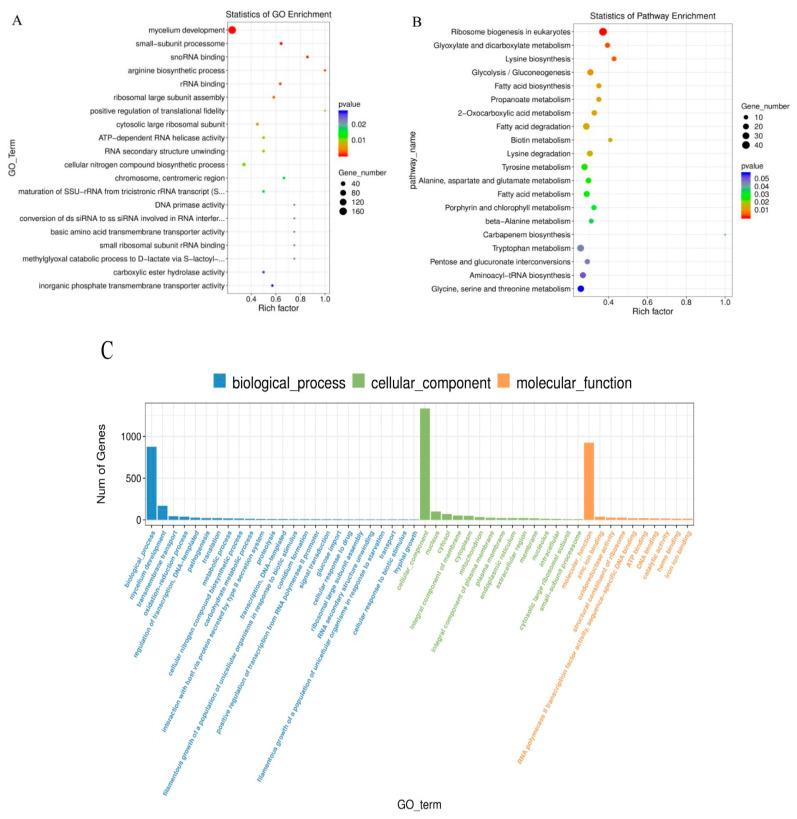
Summary of the annotations of DEGs in *M. oryzae*. (**A**) GO classifications. (**B**) Scatter plot of KEGG enrichment of DEGs. (**C**) Enrichment of GO genes in *M. oryzae* after treatment with citral (100 μg/mL) for 24 h.

**Figure 5 jof-07-01023-f005:**
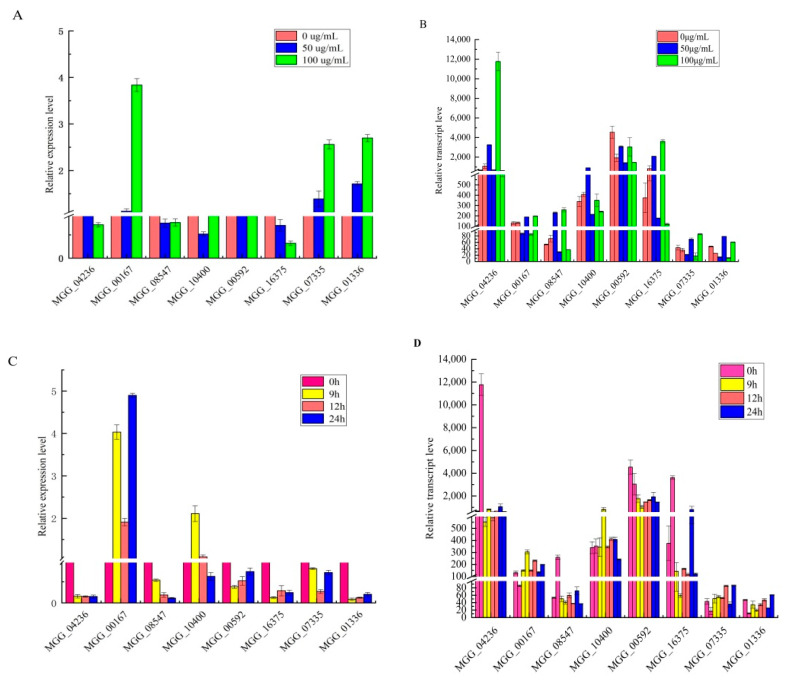
RT-qPCR verification of key genes of resistance to citral in *M. oryzae*. (**A**,**B**) The expression trend of genes as characterized by qPCR was consistent with that of RNA-seq analysis after 9, 12, and 24 h of treatment with citral (100 μg/mL). (**C**,**D**) The expression trend of genes as characterized by qPCR was consistent with that of the RNA-seq analysis after treatment with 0, 50, and 100 μg/mL of citral for 24 h. All values are based on three technical repeats and presented as means ± SD. Different characters indicate a statistically significant difference at *p* < 0.05 by *t*-test.

**Figure 6 jof-07-01023-f006:**
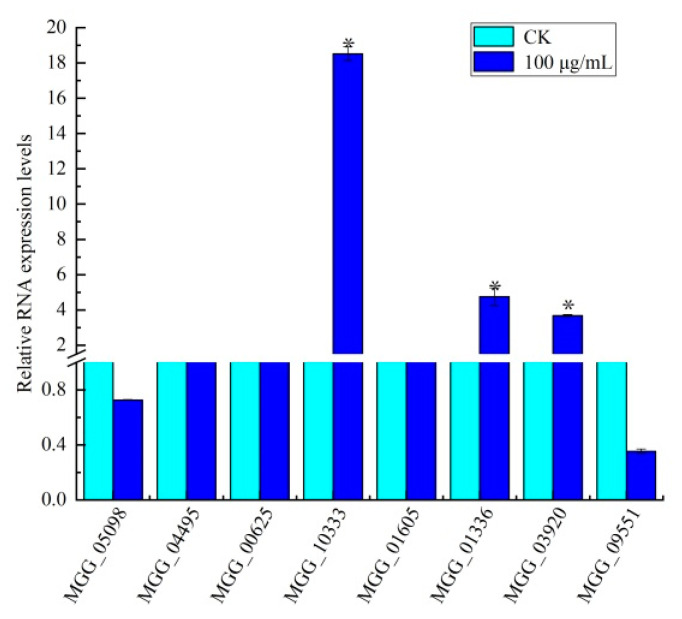
RT-qPCR analysis of key genes in the amino and nucleotide sugar pathways in response to citral treatment of *M. oryzae* after 24 h of treatment with citral (100 μg/mL). Significance is denoted by *.

**Figure 7 jof-07-01023-f007:**
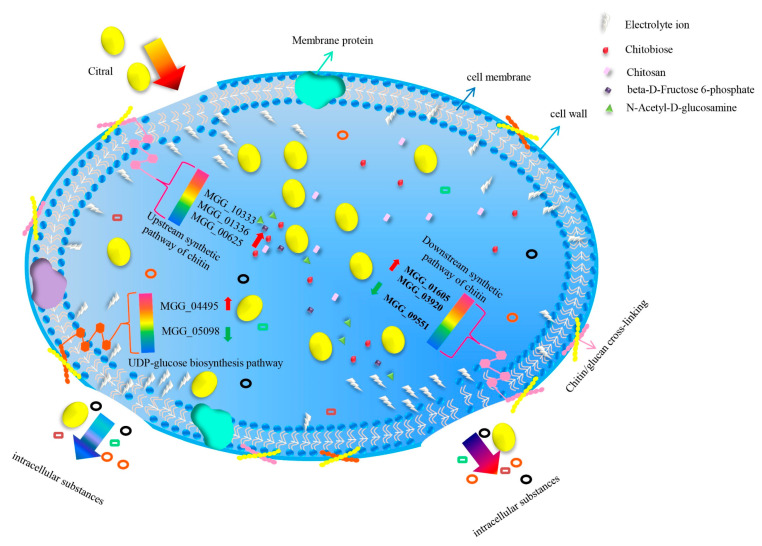
Model diagram illustrating the citral’s mechanism against the cell wall of *M. oryzae*.

**Table 1 jof-07-01023-t001:** Primer design for RT-qPCR.

Gene	Primer Name	Forward Primer (5′–3′)	Reverse Primer (5′–3′)
actin	*MGG_*03982	CTGGCACCGTCGTCGATGAAG	AAGGTCCGCTCTCGTCGTACTC
cell-wall glucanosyltransferase	*MGG_*00592	TCACCGCCATCTACGAGTCGATC	TGTCACCATCCTTGCTGCTGTTG
thioredoxin	*MGG_*04236	AATCCGGTCGCTTGCATCATCG	TCCTCTGGTGGTTGTTGCTGATTG
aldehyde reductase	*MGG_*16375	GCGGCGAGGTTGACGTGATC	CTTGGTGGCAGGCAGGTTCATG
uncharacterized protein	*MGG_*00167	CCTGACCTGACGCACTTCTACAAG	TCCTCCGAGTTCCACCAGTAAGAC
cAMP-dependent protein kinase regulatory subunit	*MGG_*07335	GCGGCTTCACCAGTCCATTCG	GATTCCGTCGGCCTCCTCCTC
CAMK/CAMK1 protein kinase	*MGG_*08547	CCAGCCATCCTCAACAACCTCAAC	TACTCGCCACTCTCATCGCTATCG
GPI-anchored cell wall β-1,3-endoglucanase EglC	*MGG_*10400	CGCTTCTACGACGGTCTGAACAAG	GCCACAGCCTGGTTGATCTGC
Bacteriodes thetaiotaomicron symbiotic chitinase	*MGG_*01336	GGCTGGCTATTGCGGTACATCTG	GCGGCAACGGCTTGGTAGTAG

## Data Availability

Sequencing data are available from https://www.ncbi.nlm.nih.gov/genome/?term=Magnaporthe%20oryzae, accessed on 21 November 2021.

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
