# Peer review of "The Antifungal Effects of Citral on Magnaporthe oryzae Occur via Modulation of Chitin Content as Revealed by RNA-Seq Analysis"

_jof, 2021, doi:10.3390/jof7121023_

Round 1

Reviewer 1 Report

My details of revision in the attached pdf file

Author Response

October 25,2021

Journal of Fungi

Manuscript Number: jof-1433869

Dear Panel Judges

Thanks very much for your attention and consideration to our manuscript! In revised manuscript, we have taken into consideration every comment/question raised by the reviewers, and made corresponding changes. Our point-by-point responses to the reviewers' comments are summarized in an itemized list.

We hope that this revision can be considered for publication by Journal of Fungi. Please feel free to contact me at +86 15185148063 by phone, or [email protected].

Thank you very much for your attention and consideration.

Best wishes,

Rong-Yu Li

Institute of Crop Protection, Guizhou University

Jiaxiunan Road 515, Huaxi District, Guiyang, 550025,Guizhou, China

Phone: +86-15185148063

Response to Reviewer 1 Comments

Point 1: please define all abberviations in the text(DEGS).

Response 1:Thanks very much for your attention to our paper! I have marked text in red. The following was a detailed explanation:

Abstract: The natural product citral has previously been demonstrated to possess antifungal activity against Magnaporthe oryzae. The purpose of this study was to screen and annotate genes that were differentially expressed (DEGs) in M. oryzae after treatment with citral using RNA sequencing (RNA-seq). Then samples were reprepared for quantitative real-time PCR(RT-qPCR) analysis verification of RNA-seq data. The results showed that 649 DEGs in M. oryzae were significantly affected after treatment with citral (100 μg/mL) for 24 h. Kyoto Encyclopedia of Genes and Genomes(KEGG) and Gene Ontology(GO) analysis showed that DEGs were mainly enriched in amino sugar and nucleotide sugar metabolic pathways, including chitin synthesis pathway and UDP sugar synthesis pathway. Results showed also that through RT-qPCR analysis the chitin present in M. oryzae might be degraded to chitosan, chitobiose, N-acetyl-D-glucosamine, and β-D-fructose-6-phosphate following treatment with citral. Chitin degradation was indicated by damaged cell-wall integrity. While the UDP glucose synthesis pathway was involved in glycolysis and gluconeogenesis, providing precursors for the synthesis of polysaccharides. Galactose-1-phosphate uridylyltransferase involved in the regulation of UDP-α-D-galactose and α-D-galactose-1-phosphate was downregulated, which would result in the inhibition of UDP glucose (UDP-Glc) synthesis, a reduction in cell-wall glucan content, and the destruction of cell-wall integrity.

Point 2: “KEGG pathway analysis showed that …”in the begining of the phrase not use abbreviations.

Response 2: Thanks very much for your attention and consideration to our manuscript!

I had marked the text in red. The following was a detailed explanation:

Kyoto Encyclopedia of Genes and Genomes pathway analysis showed that DEGs were enriched in the amino sugar and nucleotide sugar metabolism pathways of M. oryzae, which are related to chitin formation.

Point 3: “Many plants have antifungal activities,...”cite the following paper:

  1. Elshafie H.S. and Camele I. 2017. Biomed Res. Int. Volume 2017, Article ID 9268468, 14 pages. DOI: https://doi.org/10.1155/2017/9268468
  2. Elshafie H.S., et al 2018. Europ. Food Res. Technol. 244, 1675–1682. DOI: https://doi.org/10.1007/s00217-018-3080-x.

Response 3:Thanks very much for your attention to our paper!

I had marked the text in red. The following was a detailed explanation:

  • straight matter

Many plants have antifungal activities, which are related to their antifungal constituents, including alkaloids, terpenes, polysaccharide, esters, ketones, and quinones [10–12].

  • References
  1. Elshafie H.S. and Camele I. An Overview of the Biological Effects of Some Mediterranean Essential Oils on Human Health. Biomed Res. Int. 2017, 2017, 9268468.
  2. Elshafie H.S.; Aliberti. L.; Amato. M.; Feo. V.D.; Camele. I. Chemical composition and antimicrobial activity of chia (Salvia hispanica L.) essential oil. Europ. Food Res. Technol. 2018, 244, 1675–1682.

Point 4: “Citral may act by disrupting the cell-wall,...”cite the following paper:

Elshafie H.S., et al. 2017. Molecules 22 (1435), 1-16. IF: 2.86. DOI: 10.3390/molecules22091435.

Response 4:Thanks very much for your attention to our paper!

I had marked the text in red. The following was a detailed explanation:

  • straight matter

Citral may act by disrupting the cell-wall integrity and membrane permeability of M. oryzae, thus resulting in physiological changes and cytotoxicity[14].

  • References
  1. Elshafie H.S.; Armentano. M.F.; Carmosino. M.; Bufo. S.A; Feo. V.D.; Camele. I. Cytotoxic Activity of Origanum Vulgare L. on Hepatocellular Carcinoma cell Line HepG2 and Evaluation of its Biological Activity. Molecules. 2017, 22, 1435-1451.

Point 5: “For KEGG analysis, as shown in Figure 4B, a total of 4073 annotated genes …”rewrite this phrase again, it is not clear.

Response 5: Thanks very much for your attention and consideration to our manuscript!

I had marked the text in red. The following was a detailed explanation:

For KEGG analysis, a total of 4,073 annotated genes were used for mapping in M. oryzae, the main active pathways involved were: Ribosome biogenesis in Eukaryotes, Glyoxylate and dicarboxylate metabolism, Lysine degradation, Glycolysis.

Point 6: “However, the eight DEGs were related to the chitin formation…”do you have a reasonable reference confirming this information?

Response 6:Thanks very much for your attention to our paper!

I had marked the text in red. The following was a detailed explanation:

According to Supplementary Figure S3, there are nine related enzyme nodes differentially expressed in amino sugar and nucleotide sugar metabolism, among which five enzyme nodes are associated with chitin synthesis pathway. There are three enzyme nodes associated with the UDP glucose synthesis Pathway. So the eight DEGs were related to the chitin formation and UDP glucose synthesis pathways according to the enzyme node enrichment analysis.

Supplementary Fig. S3 The genes differentially expressed in Amino sugar and uncleotide sugar metabolism pathways of M. oryzae treated with 100 ug/mL citral at 24h. Red represents significantly upregulated transcripts, green represents significantly downregulated transcripts, hollow circles represent small molecular compounds, solid arrows indicate the direction of biochemical reactions, and dotted arrows link other relevant metabolic pathways.

Point 7: “Figure 6. RT-qPCR analysis of key genes in amino…”where is the figure 6?

Response 7:Thanks very much for your attention to our paper!

I had marked the text in red. The following was a detailed explanation:

Figure 6. RT-qPCR analysis of key genes in amino and nucleotide sugar pathways in response to citral treatment of M. oryzae after 24 h treatment with citral (100 μg/mL).

Point 8: “…human health [24–26].” cite the following papers:

  1. Camele I., et al 2019. Bacillus mojavensis: Biofilm formation and biochemical investigation of its bioactive metabolites. J. Biol. Res. 92 (8296), 39-45. DOI: 10.4081/jbr.2019.8296.
  2. Della Pepa T., et al 2019. Antimicrobial and phytotoxic activity of Origanum heracleoticum and O. majorana essential oils growing in Cilento (Southern Italy). Molecules 24, 2576, 1-16. DOI: 10.3390/molecules24142576.
  3. Gruľová, D., et al 2020. Thymol Chemotype Origanum vulgare L. Essential Oil as a Potential Selective Bio-Based Herbicide on Monocot Plant Species. Molecules 25 (3), 595. DOI: 10.3390/molecules25030595.
  4. Elshafie H.S., et al 2020. Mycoremediation effect of Trichoderma harzianum strain T22 combined with ozonation in diesel-contaminated sand. Chemosphere 252 , 126597. DOI: 10.1016/j.chemosphere.2020.126597.

Response 8:Thanks very much for your attention to our paper!

I had marked the text in red. The following was a detailed explanation:

  • straight matter

it may also potentially harm the environment and human health [26–29].

  • References
  1. Camele. I.; Elshafie H.S.; Caputo. L.; Sakr. S.H.; Feo. V.D. Bacillus mojavensis: biofilm formation and biochemical investigationo of its bioactive metabolites. J. Biol. Res.. 2019, 92, 39-45.
  2. Pepa. T.D.; Elshafie H.S.; Capasso. R.; Feo. V.D.; Camele. I.; Nazzaro. F.; et al. Antimicrobial and phytotoxic activity of Origanum heracleoticum and O. majorana essential oils growing in Cilento (Southern Italy). Molecules. 2019, 24, 1-16.
  3. Gruľová. D.; Caputo. L.; Elshafie H.S.; Baranová. B.; Martino. L.D.; Sedlák. V.; et al. Thymol Chemotype Origanum vulgare L. Essential Oil as a Potential Selective Bio-Based Herbicide on Monocot Plant Species. Molecules. 2020, 25, 595-612.
  4. Elshafie H.S.; Camele. I.; Sofo. A.; Mazzone. G.; aivano. M.; Masi. S.; et al. Mycoremediation effect of Trichoderma harzianum strain T22 combined with ozonation in diesel-contaminated sand. Chemosphere. 2020, 252, 1-8.

Point 9: “KEGG pathway analysis showed that citral could…”how did you confirm this action of citral against chitin, just by demonstrating the decreasing in the content of chitin?

Response 9:Thanks very much for your attention to our paper!

The following was a detailed explanation:

My basis is based on the preliminary experimental results of our research group. our previous research results showed that citral had a strong antibacterial activity against M. oryzae. The results of enzyme activity and proteomics determination showed that citral may increase the chitinase activity of M. oryzae, destroy the structure of cell wall and inhibit the growth of M. oryzae. Therefore, our study via that the RNA-Seq Analysis and real-time fluorescence quantitative verification that the citral could affect chitin synthesis and UDP glucose synthesis in the amino sugar and nucleotide metabolic pathways of M. oryzae. RT-qPCR analysis the chitin present in M. oryzae might be degraded to chitosan, chitobiose, N-acetyl-D-glucosamine, and β-D-fructose-6-phosphate following treatment with citral. Chitin degradation was indicated by damaged cell-wall integrity. While the UDP glucose synthesis pathway was involved in glycolysis and gluconeogenesis, providing precursors for the synthesis of polysaccharides. Galactose-1-phosphate uridylyltransferase involved in the regulation of UDP-α-D-galactose and α-D-galactose-1-phosphate was downregulated, which would result in the inhibition of UDP glucose (UDP-Glc) synthesis, a reduction in cell-wall glucan content, and the destruction of cell-wall integrity. At present, our research group is constructing silencing vectors for the target genes were screened in this study, and observing the genotype expression and other related work in the later stage.

The following is the results of this research group and RT-qPCR analysis of key genes in amino and nucleotide sugar pathways in response to citral treatment of M. oryzae after 24 h treatment with citral (100 μg/mL) in Figure 6:

  • Li, R.Y.; Wu, X.M.; Yin, X.H.; Liang, J.N.; LI, M.The Natural Product Citral Can Cause Significant Damage to the Hyphal Cell Walls of Magnaporthe grisea. Molecules 2014, 19, 10279-10290.
  • Li, R.Y.; Wu, X.M.; Yin, X.H.; Long, Y.H.; Li, M. Naturally produced citral can significantly inhibit normal physiology and induce cytotoxicity on Magnaporthe grisea. Pestic Biochem Physiol 2015, 118, 19-25.
  • Zhao QJ.; Ding, Y.; Song, X.C.; Liu, S.J.; Li, M.; LI, R.Y.; et al. Proteomic analysis reveals that naturally produced citral can significantly disturb physiological and metabolic processes in the rice blast fungus Magnaporthe oryzae. Pestic Biochem Physiol 2021, 175, 104835.

Figure 6. RT-qPCR analysis of key genes in amino and nucleotide sugar pathways in response to citral treatment of M. oryzae after 24 h treatment with citral (100 μg/mL).

Point 10: “The RNA-seq technique was used to analyze the changes in…”this phrase should not be in the conclusion.

Response 10:Thanks very much for your attention to our paper!

I had already deleted it The following was a detailed explanation:

RNA-seq analysis identified the effects of citral concentration on M.oryzae were analyzed. The number of DEGs in M. oryzae reached the maximum value after treatment with citral (100 μg/mL) for 24 h. Most DEGs were enriched in biological processes related to mycelium development and conidia formation. KEGG pathway analysis showed that genes in the amino sugar and nucleotide sugar metabolism pathways of M. oryzae were differentially expressed following treatment with citral, and these pathways are related to chitin formation. In the chitin synthesis pathway of M. oryzae, chitin synthase gene expression was downregulated, resulting in an inhibition of chitin synthesis. Moreover, upregulation of chitinase gene expression can lead to an increase in chitin degradation, thus resulting in a decrease in chitin content and destruction of the cell-wall structure in M. oryzae. In the UDP glucose synthesis pathway of M. oryzae, UDP glucose synthesis was inhibited, which could reduce the cell-wall glucan content and disrupt the cell-wall integrity. This study further revealed that citral had excellent antifungal activity against M. oryzae by affecting the content of chitin, thus providing a scientific basis for elucidating its mode of action. At present, our research group is constructing silencing vectors for the target genes were screened in this study, and observing the genotype expression and other related work in the later stage.

Reviewer 2 Report

Comments and suggestions to the authors

The manuscript entitled “The Antifungal Effects of Citral on Magnaporthe oryzae Occur via Modulation of Chitin Content as Revealed by RNA-Seq Analysis” Manuscript ID: jof-1433869, by Song et al., provides insight on the potential molecular mechanism(s) behind the antifungal activity of the citrus-derived terpenoid, citral, against the phytopathogenic fungus Magnaporthe oryzae. In the current study, the authors attempted to screen and annotate the gene targeted by citral and responsible for its antifungal activity against M. oryzae using RNA sequencing (RNA-seq). Although the idea is interesting, the manuscript is not well-planned and poorly visualized and written as well. In addition, there is no experimental evidence that supports their conclusion. Also, the data do not fully support the model.

Major concerns:-

  1. Firstly, my main concern is the high plagiarism (>30%) throughout the manuscript. Some sections should paraphrase before the acceptance of the manuscript for publishing particularly the “Materials and Methods” section. Several sections have been typically copied from another paper previously published in PLOS one entitled “De novo leaf and root transcriptome analysis to identify putative genes involved in triterpenoid saponins biosynthesis in Hedera helix” by Sun, et al., https://doi.org/10.1371/journal.pone.0182243. PLEASE, consider paraphrasing these paragraphs. This issue should be carefully revised throughout the manuscript.
  2. Secondly, the originality/ Novelty of the work is weak, fuggy, and highly speculative. For example, in the title the authors claimed that “The antifungal effects of citral on oryzae occur via modulation of chitin content”, however, I did not find any quantitative, or even qualitative work, for chitin content throughout the manuscript. All that I see is only some differentially expressed genes (DEGs), some of them are related to chitin synthesis.

Specific comments are shown below:-

  1. Abstract: Although the abstract is easy-to-read, however, it is too long and descriptive. According to “Instructions for Authors” of JoF, the abstract should be a total of about 200 words maximum (for more information, please see https://www.mdpi.com/journal/jof/instructions). Moreover, it does not present the main conclusion of the work. For example, the last third of the abstract (lines 28-35) is very speculative without strong experimental evidence. I highly recommend rephrasing the abstract to articulate more clearly the novelty of this work. It needs to be more informative rather than descriptive.
  2. Introduction: in contrast with the abstract, the introduction is very short and not sufficiently The introduction should briefly place the study in a broad context and highlight why it is important. It should define the purpose of the work and its significance, including specific hypotheses being tested. I highly recommend rewriting the introduction section emphasizing the objectives of the study.
  3. Materials and methods: as I mentioned above, the ‘materials and methods’ section has high plagiarism with published work. PLEASE, consider paraphrasing these paragraphs. This issue should be carefully revised throughout the manuscript.
  4. Results: Throughout the results section, the presentation of the figures is very poor and must be enhanced before publishing. For example, the authors should use the same font and font size (at least 8 pt) in all figures throughout the manuscript. Moreover, all figures should have a short explanatory title and caption. However, most of the figures’ captions are not sufficient. For example:-
  • Figure 1: use the same font and font size, move the panel lettering (A, B, and C) to the upper left corner of each panel.
  • Figure 2: use the same font and font size, add the panel lettering (A, B, C, …..etc.) to the upper left corner of each panel. Moreover, don’t italicize the figure caption.
  • Figure 3: use the same font and font size, add the panel lettering (A, B, C, …..etc.) to the upper left corner of each panel.
  • Figure 4: all the figure must be redone since it is not readable with very low quality.
  • Figure 5: use the same font and font size, explain what are the bars (means or something else) and error bars (SE, SD, or something else).
  • Figure 6: is MISSING.
  • Figure 7: is poorly designed and not related to the results section.
  1. Discussion: the discussion section is the weakest part of this manuscript. Discussion is very short and insufficient. Discussion must be rewritten. I believe the hypothetical model presented in Figure 7 and its related text fit the ‘Discussion’ section better than ‘Results’ and it will be very helpful and supportive to the conclusion of this study.
  2. Supplementary materials: although the authors mentioned that contains four tables and three figures, I can not find the supporting file anywhere neither within the manuscript nor online on the JoF website. All supplementary materials must be submitted within the revised version of this manuscript.

Finally, Although the language used in the manuscript is easy to follow and understand, however, the manuscript should be carefully and deeply revised for grammar and English use, since some other mistakes were found throughout the whole paper.

Author Response

October 25,2021

Journal of Fungi

Manuscript Number: jof-1433869

Dear Panel Judges

Thanks very much for your attention and consideration to our manuscript! In revised manuscript, we have taken into consideration every comment/question raised by the reviewers, and made corresponding changes. Our point-by-point responses to the reviewers' comments are summarized in an itemized list.

We hope that this revision can be considered for publication by Journal of Fungi. Please feel free to contact me at +86 15185148063 by phone, or [email protected].

Thank you very much for your attention and consideration.

Best wishes,

Rong-Yu Li

Institute of Crop Protection, Guizhou University

Jiaxiunan Road 515, Huaxi District, Guiyang, 550025,Guizhou, China

Phone: +86-15185148063

Response to Reviewer 2 Comments

Point 1:Abstract: Although the abstract is easy-to-read, however, it is too long and descriptive. According to “Instructions for Authors” of JoF, the abstract should be a total of about 200 words maximum (for more information, please see https://www.mdpi.com/journal/jof/instructions). Moreover, it does not present the main conclusion of the work. For example, the last third of the abstract (lines 28-35) is very speculative without strong experimental evidence. I highly recommend rephrasing the abstract to articulate more clearly the novelty of this work. It needs to be more informative rather than descriptive.

Response 1: Thank you very much for valuable comments of reviewer. I had already amended it, and the following was a detailed explanation:

Abstract: The natural product citral has previously been demonstrated to possess antifungal activity against Magnaporthe oryzae. The purpose of this study was to screen and annotate genes that were differentially expressed (DEGs) in M. oryzae after treatment with citral using RNA sequencing (RNA-seq). Then samples were reprepared for quantitative real-time PCR(RT-qPCR) analysis verification of RNA-seq data. The results showed that 649 DEGs in M. oryzae were significantly affected after treatment with citral (100 μg/mL) for 24 h. Kyoto Encyclopedia of Genes and Genomes(KEGG) and Gene Ontology(GO) analysis showed that the DEGs were mainly enriched in amino sugar and nucleotide sugar metabolic pathways, including chitin synthesis pathway and UDP sugar synthesis pathway. Results showed also that through RT-qPCR analysis the chitin present in M. oryzae might be degraded to chitosan, chitobiose, N-acetyl-D-glucosamine, and β-D-fructose-6-phosphate following treatment with citral. Chitin degradation is indicated by damaged cell-wall integrity. In addition, the UDP glucose synthesis pathway is involved in glycolysis and gluconeogenesis, providing precursors for the synthesis of polysaccharides. Galactose-1-phosphate uridylyltransferase involved in the regulation of UDP-α-D-galactose and α-D-galactose-1-phosphate, was downregulated, which would result in the inhibition of UDP glucose (UDP-Glc) synthesis, a reduction in cell-wall glucan content, and the destruction of cell-wall integrity.

Point 2: Introduction: in contrast with the abstract, the introduction is very short and not sufficiently The introduction should briefly place the study in a broad context and highlight why it is important. It should define the purpose of the work and its significance, including specific hypotheses being tested. I highly recommend rewriting the introduction section emphasizing the objectives of the study.

Response 2: Author would like to thank constructive comments of reviewer. I had amended it. The following was a detailed explanation:

Introduction

The fungus M. oryzae causes rice blast, which is the most widespread and serious disease affecting rice [1,2] and one of the most destructive and economically harmful diseases affecting rice production worldwide [3,4]. At present, the methods of disease-resistant variety breeding and chemical control were mainly used for prevention and control of M. oryzae in China[5]. Due to shortcomings of traditional breeding of disease-resistant varieties, such as long breeding cycle, easy to loss resistance, high cost of research and development of chemical pesticides, pesticide residues and drug resistance[6], it was urgented to move away from the traditional excessive dependence on chemical pesticides during crop management. Under the premise of stabilizing production, intensive efforts were needed to develop biological pesticide technology with low toxicity and low residue characteristics.

The discovery of antifungal and germicidal compounds from plants has led to the efficient creation of new pollution-free pesticides [7–9]. Many plants have antifungal activities, which were related to their antifungal constituents, including alkaloids, terpenes, polysaccharide, esters, ketones, and quinones [10–12], which were lemongrass, cilantro, cassia, cinnamon[13]. Especially, Citral is an open-chain monoterpene extracted from Litsea cubeba oil, which has good antifungal, insecticidal, and antioxidant activities [14].

Our previous studied demonstrated that citral has significant biological activity against M. oryzae in vitro and in vivo, suggesting its potential application in controlling rice blast [15,16]. Citral may act by disrupting the cell-wall integrity and membrane permeability of M. oryzae, thus resulting in physiological changes and cytotoxicity[17]. Importantly, the inhibitory effect of citral on M. oryzae appears to be associated with its effects on reducing sugar, soluble protein, pyruvate, and malondialdehyde contents, as well as chitinase activity, in mycelia [16,18]. However, the gene targeted by citral and thus responsible for its activity on M. oryzae has not been identified. Recent advances in high-throughput genomic technology, including transcriptomics, have provided a new method for studying fungal growth, pathogenesis, and genes involved in resistance [19–21]. Thus, we decided to study the gene expression profile of citral-treated M. oryzae to gain some insight into its mode of action. In this research, The objectives were (i) to screen and annotate the differentially expressed genes (DEGs) of M. oryzae after treatment with citral using RNA-seq, (ii) to ascertain the expression profiles of these M. oryzae DEGs, and (iii) to use the information about DEGs and their expression profiles to construct a model diagram illustrating the mechanism of citral on M. oryzae. Furthermore, the samples were reprepared for real-time fluorescence quantitative verification of RNA-seq data. The inhibitory targets of citral on M. oryzae were obtained through the changes of DEGs levels and RT-qPCR analysis verification results. As far as we know, this is the first report on using RNA-seq method to study mechanism of citral on M. oryzae. Studies of closely related response genes and signaling pathways can allow elucidating the molecular mechanism of citral acting on M. oryzae, as well as provide a scientific basis for elucidating its mode of action and a valuable reference for the control of M. oryzae.

Point 3: Materials and methods: as I mentioned above, the ‘materials and methods’ section has high plagiarism with published work. PLEASE, consider paraphrasing these paragraphs. This issue should be carefully revised throughout the manuscript.

Response 3: The reviewer's remark is constructive.I had amended it The following was a detailed explanation:

  1. Materials and Methods

2.2. RNA Extraction and Preparation of cDNA Library

Total RNA was isolated using TRIzol reagent (Invitrogen, Carlsbad, CA, USA) following the manufacturer’s instructions. The yield and purity of each RNA sample were quantified using a NanoDrop ND-1000 (NanoDrop, Wilmington, DE, USA). The RNA integrity was assessed as a function of the RNA integrity number (RIN) >7.0 ,using the Agilent 2100 bioanalyzer.

Poly(A) RNA was purified from total RNA(5 µg) using poly-T oligo-attached magnetic beads using two rounds of purification. Then the poly(A) RNA was fragmented into small pieces using divalent cations under high temperature. Then the cleaved RNA fragments were reverse-transcribed to create the cDNA, which were next used to synthesise U-labeled second-stranded DNAs with E. coli DNA polymerase I, RNase H and dUTP A-bases were added to the blunt ends of each strand, preparing them for T-base overhang adapter ligation. Single or dual index adapters were ligated to the fragments, and size selection was performed with AMPureXP beads. After the heat-labile UDG enzyme treatment of the U-labeled second-stranded DNAs, the ligated products were amplified by PCR using the following conditions: initial denaturation at 95℃ for 3 min; followed by 8 cycles of denaturation at 98℃ for 15 s, annealing at 60℃ for 15 s, and extension at 72℃ for 30 s; and a final extension at 72℃ for 5 min. The average insert size for the final cDNA library was 300 bp (±50 bp). At last, the 150bp paired-end sequencing were performed on an Illumina Hiseq 4000 (LC Bio, China). The bioinformatics analysis pipeline is shown in Supplementary Figure S1.

2.3. Sequence Assembly and Annotation

Raw data generated by sequencing needs to be preprocessed. Cutadapt were used to filter out unqualified sequences and obtain valid data before further analysis[23]. Reads were removed with adaptor, containing N(N indicates that base information cannot be determined) were removed of more than 5%. Low quality Reads (more than 20% of the total read with a mass value of Q≤10) were removed. The original sequencing amount, effective sequencing amount, Q20, Q30 and GC contents were counted and comprehensively evaluated. Then, sequence quality was verified using FastQC (http://www.bioinformatics.babraham.ac.uk/ projects/ fastqc/). The known annotation information in the species database (https://www.ncbi.nlm.nih.gov/genome/?term= Magnaporthe%20oryzae) was analysed, including chromosome number, gene number, transcript number, GO annotation(http://geneontology.org) and KEGG annotation(http://www.kegg.jp/kegg).

2.4. Gene Expression Analysis

The algorithm feature Counts version 1.5.0-p3 was used for FPKM calculation of each gene, according to the length of the gene and the read count mapped to this gene [24]. Expression levels of different samples of M. oryzae treated with citral by calculating a whole-genome model [25,26]. To use edgeR for finished StringTie assembly and quantitative genetic variance analysis (significant difference threshold for | log2 foldchange |≥1, p<0.05) [27]. In the end,the volcano figure, GO and KEGG enrichment analyses were performed on the DEGs.

2.5. Quantitative Real-Time PCR Analysis

The total RNA for RT-qPCR analysis was extracted using a 2× SG Fast qPCR Master Mix (Sangon, Shanghai, China), and 1.0 μg of RNA was used for reverse transcription with a M-MuLV First-Strand cDNA Synthesis Kit (Sangon, Shanghai, China) in a 20 μL reaction volume according to the manufacturer’s instructions. Gene-specific primer pairs were designed using the Sangon primer design and synthesis online platform (https://www.sangon.com/newPrimerDesign) on the basis of the transcript sequence and synthesized by Sangon Biotech Co., Ltd (Shanghai, China) (Table 1). The actin gene was used as an internal control [28]. The PCR was performed using SYBR Green's method and a Bio-Rad CFX96 real-time PCR system (Bio-Rad, CA, USA) with a total of 96 well plates, each sample was set up with three biological replicates. The final volume for each reaction was 20 μL with the following components: 2 μL cDNA template (1 ng/μL), 10 μL 2xSG Fast qPCR Master Mix (Sangon, Shanghai, China), 0.4 μL forward primer (200 nM), 0.4 μL reverse primer (200 nM), and 7.2 μL PCR-grade water. The reaction was conducted under the following conditions: 95˚C for 3 min, followed by 40 cycles of denaturation at 95℃ for 10 s and annealing/extension at 56℃for 30 s. The melting curve was obtained by heating the amplicon from 65℃ to 95℃ at increments of 0.5℃ per 5 s. The relative quantification of gene expression was computed using the 2−ΔΔCt method.

Point 4: Results: Throughout the results section, the presentation of the figures is very poor and must be enhanced before publishing. For example, the authors should use the same font and font size (at least 8 pt) in all figures throughout the manuscript. Moreover, all figures should have a short explanatory title and caption. However, most of the figures’ captions are not sufficient. For example:-

    Figure 1: use the same font and font size, move the panel lettering (A, B, and C) to the upper left corner of each panel.

    Figure 2: use the same font and font size, add the panel lettering (A, B, C, …..etc.) to the upper left corner of each panel. Moreover, don’t italicize the figure caption.

    Figure 3: use the same font and font size, add the panel lettering (A, B, C, …..etc.) to the upper left corner of each panel.

    Figure 4: all the figure must be redone since it is not readable with very low quality.

    Figure 5: use the same font and font size, explain what are the bars (means or something else) and error bars (SE, SD, or something else).

    Figure 6: is MISSING.

Figure 7: is poorly designed and not related to the results section.

Response 4: Thanks very much for your attention to our paper! I had amended it. The following was a detailed explanation:

Figure 1. Venn diagram of DEGs in the evaluated libraries. (A) Venn map of all DEGs in M. oryzae after treatment with citral (50 μg/mL) for the indicated times (h). (B) Venn map of all DEGs in M. oryzae after treatment with citral (100 μg/mL) for the indicated times (h). (C) Venn map of specific DEGs in M. oryzae after treatment with citral for 24 h.

Figure 2. Volcanic map of DEGs in M. oryzae after treatment with citral at different concentrations. Red represents upregulated significantly differentially expressed genes, blue represents downregulated significantly differentially expressed genes, and gray dots represent non significantly differentially expressed genes.

Figure 3. Volcanic map analysis of DEGs in M. oryzae after treatment with citral for 24 h. Red rep-resents upregulated significantly differentially expressed genes, blue represents downregulated significantly differentially expressed genes, and gray dots represent non significantly differentially expressed genes.

Figure 4. Summary of the annotations of DEGs in M. oryzae. (A) GO classifications. (B) Scatter plot of KEGG enrichment of DEGs. (C) Enrichment of GO genes in M. oryzae after treatment with citral (100 μg/mL) for 24 h.

Figure 5. RT-qPCR verification of key genes of resistance to citral in M. oryzae. (A,B) The expression trend of genes as characterized by qPCR is consistent with that of RNA-seq analysis after 9, 12, and 24 h treatment with citral (100 μg/mL). (C,D) The expression trend of genes as characterized by qPCR is consistent with that of RNA-seq analysis after treatment with 0, 50, and 100 μg/mL citral for 24 h. All values are based on three technical repeats and presented as means±SD. Different characters indicate a statistically significant difference at p< 0.05 by t-test.

Figure 6. RT-qPCR analysis of key genes in amino and nucleotide sugar pathways in response to citral treatment of M. oryzae after 24 h treatment with citral (100 μg/mL).

Figure 7. Model diagram illustrating the mechanism of citral against the M. oryzae cell wall.

Point 5: Discussion: the discussion section is the weakest part of this manuscript. Discussion is very short and insufficient. Discussion must be rewritten. I believe the hypothetical model presented in Figure 7 and its related text fit the ‘Discussion’ section better than ‘Results’ and it will be very helpful and supportive to the conclusion of this study.

Response 5: Author would like to thank constructive comments of reviewer. I had already amended it. The following was a detailed explanation:

  1. Discussion
  2. oryzae causes huge economic losses in global agriculture. Although the wide-spread use of traditional fungicides can reduce crop loss, it may also potentially harm the environment and human health [29-32]. Therefore, it is very important to develop safe and green natural pesticide products. At present, microorganisms, microbial products, and biological fertilizers have been developed as biological control agents [33].

Transcriptomics technology allows analyzing the changes in RNA levels in micro-organisms under different environments at a molecular level from a macro perspective [34–37]. For example, on the basis of the results of RNA-seq, the effects of sodium pheophorbide a (SPA) treatment on the expression of three cell-wall-degrading en-zyme-related genes in Pestalotiopsis neglecta hypha were detected using qRT-PCR, thereby revealing the molecular mechanism underlying the action of SPA on P. neglecta. SPA in-hibited TCA-related enzyme activity and the expression of three cell-wall-degrading enzyme-related genes [38]. The genome and transcriptome of the pine saprophyte Ophiostoma piceae) comparison with the genome and transcriptomes of the mountain pine beetle-associated pathogen Grosmannia clavigera differences between a pathogen that found a major class of pine tree defense compounds[39]. Camptothecin (CPT) is a botanical compound primarily isolated from Camptotheca acuminata Decne, which is an important medicinal plant in China. CPT is a promising lead compound for developing fungicides against rice blast, as it may bind to the DNA topoisomerase I complex of M. oryzae, thus affecting translation and carbohydrate metabolism/energy metabolism, leading to cell death [40]. Our group previously studied citral extracted from Litsea cubeba fruit, which had properties consistent with those of leaf oil extracted from Eucalyptus and showed a good inhibitory effect on M. oryzae growth [41]. In addition, they have research found the significant antibacterial activity of citral in Aspergillus ochraceus, Penicillium, Penicillium roqueforti, and so on[42-44]. In this study, the transcriptome analysis method based on RNA-seq was used to analyze the RNA levels of genes in M. oryzae treated with citral. The results show that citral might act in disrupting cell-wall integrity and the cytoplasmic membrane by causing changes in membrane permeability, leading to a loss of intracellular sub-stances and the introduction of wrinkles and depressions on the cell surface.

Chitin is a β-1,4-linked polymer of N-acetylglucosamine and a major component of the fungal cell wall [45,46]. Fungi can successfully invade the host by destroying the conserved chitin in the fungal cell wall. Chitin hydrolase CcCti1 inhibited conidial germination and appressoria formation of M. oryzae, thereby affecting its normal growth [47]. In this study, most DEGs were enriched in biological processes related to mycelium development and conidia formation. The polar germination of conidia and the growth of infected hyphae of M. oryzae were related to dynamic changes in the M. oryzae cell wall. It was found that the expressions of MGG_10333 (class III chitinase), MGG_01336 (Bacteriodes thetaiotaomicron symbiotic chitinase), and MGG_00625 (glucosamine-6-phosphate isomerase) were upregulated in M. oryzae after treatment with citral. Thus, the chitin present in M. oryzae might be degraded to chitosan, chitobiose, N-acetyl-D-glucosamine, and β-D-fructose-6-phosphate following treatment with citral. The expression of FAD-binding domain-containing protein MGG_01605 was upregulated in M. oryzae, and this gene was found downstream of the chitin synthesis pathway and involved in the regulation of chitin. The expression of inositolphosphorylceramide-B C-26 hydroxylase MGG_03920 was upregulated, involved in ferrocytochrome B5 synthesis. Furthermore, the expression of chitin synthase MGG_09551 was downregulated, which was inhibition of chitin synthesis. Chitin degradation is indicated by damaged cell-wall integrity, as an important component of cell wall in M. oryzae [48]. KEGG pathway analysis showed that citral could affect chitin synthesis and UDP glucose synthesis in the amino sugar and nucleotide metabolic pathways of M. oryzae(Figure 7). UDP glucose is a donor in most polysaccharide synthesis pathways and the pre-cursor of glucan synthesis [49]. UDP glucose synthesis pathway is involved in glycolysis and gluconeogenesis, providing precursors for synthesis of polysaccharides. The expression of phosphoglucomutase MGG_04495 was upregulated in M. oryzae after treatment with citral, which was involved in catalyzing mutual transformation of glucose-6-phosphate and glucose-1- phosphate in UDP glucose synthesis pathway. Galactose-1-phosphate uridylyl transferase MGG_05098, involved in regulation of UDP-α-D-galactose and α-D-galactose-1-phosphate, was downregulated, which would result in inhibition of UDP glucose (UDP-Glc) synthesis, a reduction in cell-wall glucan content, and destruction of cell-wall integrity.

In conclusion, the lipophilic characteristic of citral facilitated its passage and access such that it can disrupt cell-wall integrity and the cytoplasmic membrane by causing changes in membrane permeability, leading to the loss of intracellular substances and the introduction of wrinkles and depressions on the cell surface[50]. The model diagram shows that the gene expression of chitin synthase was downregulated in the chitin formation pathway of M. oryzae after treatment with citral, leading to an inhibition of chitin synthesis. On the other hand, the gene expression of chitinase was upregulated, resulting in a reduction in chitin content in the cell wall of M. oryzae, leading to damage. However, the expressions of the glucosamine-6-phosphate deaminase and UDP glucose pyrophosphorylase genes were downregulated, resulting in an inhibition of UDP glucose synthesis and a reduction in the cell-wall glucan content, thereby disrupting the M. oryzae cell-wall integrity and inhibiting mycelial growth. This study identified the key genes related to cell-wall integrity in M. oryzae which exhibited a response to citral, further revealing its excellent antifungal ac-tivity against M. oryzae occurs via effects on chitin content, thus providing a scientific basis for elucidating its mode of action. Therefore, the regulatory molecular mechanism of transcription factors in mediating chitinase in M. oryzae following treatment with citral still need further study.

Point 6: Supplementary materials: although the authors mentioned that contains four tables and three figures, I can not find the supporting file anywhere neither within the manuscript nor online on the JoF website. All supplementary materials must be submitted within the revised version of this manuscript.

Response 6: Thanks very much for your attention to our paper! The following was a detailed explanation:

Supplementary Materials

Supplementary Fig 1. Experimental flow

Supplementary Fig. 2 Systematic clustering of differentially expressed genes (DEG) in 36 transcriptome libraries

Supplementary Fig. S3 Pathway of amino and nucleotide sμgar metabolism of M. oryzae

Supplementary Table S1. 8 genes of primer design for RT-qPCR

Supplementary Table S2. Summary of reference genetic data from processed samples

Supplementary Table S3. Comparison of transcriptome sequencing and reference genome of M. oryzae under citral stress

Supplementary Table S4. Large value distribution statistics of reference gene table for transcriptome sequencing of M. oryzae under citral 

Reviewer 3 Report

Dear authors,

In the present study, "The Antifungal Effects of Citral on Magnaporthe oryzae Occur via Modulation of Chitin Content as Revealed by RNA-Seq Analysis" by Song et al., the authors describe citral as a potential alternative to chemical pesticides against the rice blast fungus Magnaporthe oryzae. Citral possesses antifungal activity by disrupting cell-wall integrity, thus causing an increase in membrane permeability, resulting in physiological changes and cytotoxicity. The authors identified the genes targeted by citral in M. oryzae by RNAseq analysis. They identified a total of 21,502 differentially expressed genes (DEGs). Here, 2,301 DEGs were significantly affected after treatment with citral. Those genes are involved in the biosynthesis of mycelium development, conidium formation, and the transmembrane transport process. KEGG pathway analysis showed that amino sugar and nucleotide sugar metabolism pathways of M. oryzae were affected. The authors finally present a model diagram illustrating the mechanism of citral on M. oryzae, summarizing that citral treatment could reduce chitin content and cell wall damage through the inhibition of chitin synthesis. Furthermore, the differential expression of genes of the uridine diphosphate (UDP) glucose formation pathway could lead to a disruption in cell-wall integrity through inhibition of UDP glucose synthesis and a reduction in glucan content of the cell wall, thus inhibiting mycelial growth.

The manuscript deals with an exciting and essential problem: identifying effective alternatives to chemical pesticides to improve crop growth and reduce pesticides' harmful effects. Natural, biologically active compounds like citral can be a solution. Therefore, the compounds have to be characterized in detail. The present study indicates the effect of citral on an important fungal pathogen and contributes to understanding the reaction mechanism for a future application as a fungal pesticide.

The introduction nicely presents the aim of the study and describes the problem of fungal pathogens, usage of pesticides, and previous studies on citral treatments. Further, identification of differently expressed genes as a hint for citral reaction mechanism characterization is introduced. The Materials and Methods section is complete and adequately describes the used methods. Results are subsequently described in a precise manner and visualized with attractive figures. For my understanding, the authors should explicitly mention that the results rely on RNAseq data, which shows transcription profiles of genes. However, the annotation and KEGG pathways are based on pure bioinformatics results without clear experimental proof. I suggest toning down the statements/conclusions to what they are – correlations, hints, suggestions, indications. The discussion part is the weakest section of the manuscript, mainly through the kind of writing. It seems that the discussion was written by another co-author and did not fit the style of the rest of the manuscript. Paragraphs are not bridging and purely mention previous studies and literature. The part on the model diagram is, in contrast, nicely written and helps the reader to understand the complex suggested reaction mechanism. One comment is on the choice of colors in Fig. 7, which makes the model's descriptions hard to read. In addition, the conclusion section summarizes the study. It gives an outlook on current ongoing investigations, which promise the verifying experimental work to elucidate the effect of citral on the rice blast fungus. This outlook supports my criticism on toning down the "identified effect of citral" statement since lab experiments are so far missing. 

Specific comments, questions, and recommendations are implemented in the attached pdf file.

Author Response

November 10,2021

Dear Reviewer

Thanks very much for your attention and consideration to our manuscript! In revised manuscript, we have taken into consideration every comment/question raised by the reviewers, and made corresponding changes. Our point-by-point responses to the reviewers' comments are summarized in an itemized list.

We hope that this revision can be considered for publication by Journal of Fungi. Please feel free to contact me at +86 15185148063 by phone, or [email protected].

Thank you very much for your attention and consideration.

Best wishes,

Rong-Yu Li

Institute of Crop Protection, Guizhou University

Jiaxiunan Road 515, Huaxi District, Guiyang, 550025,Guizhou, China

Phone: +86-15185148063

Response to Reviewer 3 Comments Point 1: “However, the gene targeted by citral and thus responsible for its activity on M. oryzae has not been identified.” Is it hypothesized that only one gene is affected by citral?

Response 1: Thank you very much for valuable comments of reviewer. Our previous research results showed that citral had a strong antibacterial activity against M. oryzae. The results of enzyme activity and proteomics determination showed that citral may increase the chitinase activity of M. oryzae, destroy the structure of cell wall and inhibit the growth of M. oryzae. This study was to screen and annotate genes that were differentially expressed (DEGs) in M. oryzae after treatment with citral using RNA sequencing (RNA-seq). KEGG and GO analysis showed that DEGs were mainly enriched in amino sugar and nucleotide sugar metabolic pathways, including chitin synthesis pathway and UDP sugar synthesis pathway. Eight genes were screened from the chitin pathway and UDP sugar pathway, and these eight genes were exhibited a response to citral(six upregulated genes and two downregulated genes).

References are as follows:

①    Li, R.Y.; Wu, X.M.; Yin, X.H.; Liang, J.N.; LI, M.The Natural Product Citral Can Cause Significant Damage to the Hyphal Cell Walls of Magnaporthe grisea. Molecules 2014, 19, 10279-10290.

②    Li, R.Y.; Wu, X.M.; Yin, X.H.; Long, Y.H.; Li, M. Naturally produced citral can significantly inhibit normal physiology and induce cytotoxicity on Magnaporthe grisea. Pestic Biochem Physiol 2015, 118, 19-25.

③    Zhao QJ.; Ding, Y.; Song, X.C.; Liu, S.J.; Li, M.; LI, R.Y.; et al. Proteomic analysis reveals that naturally produced citral can significantly disturb physiological and metabolic processes in the rice blast fungus Magnaporthe oryzae. Pestic Biochem Physiol 2021, 175, 104835.

Point 2: “In this research, The objectives were (i) to screen and annotate…”Pay attention to upper and lower case letters.

Response 2: Author would like to thank constructive comments of reviewer. I had amended it. The following was a detailed explanation:

In this research, the objectives were (i) to screen and annotate the differentially expressed genes (DEGs) of M. oryzae after treatment with citral using RNA-seq, (ii) to ascertain the expression profiles of these M. oryzae DEGs, and (iii) to use the information about DEGs and their expression profiles to construct a model diagram illustrating the mechanism of citral on M. oryzae.

Point 3: “Eight pairs of primers were designed, and actin was used as an internal control for normalization…”I am wondering, if actin is a reasonable housekeeping gene, since structural genes are differently expressed

Response 3: The reviewer's remark is constructive. The following was a detailed explanation: actin is a reasonable housekeeping gene, the M.oryzae actin motor proteins were associated with hyphal growth and appressorial development. The actin-binding proteins were key factors in M.oryzae development and virulence through regulating actin assembly. References are as follows:

  • Li, Y.B.; Xu, R.; Liu, C.; Shen, N.; Han, L.B.; Tang, D. Magnaporthe oryzae fimbrin organizes actin networks in the hyphal tip during polar growth and pathogenesis. PLoS Pathog 2020, 16, e1008437.
  • Wang, J.; Du, Y.; Zhang, H.; Zhou, C.; Qi, Z.; Zheng, X.; et al. The actin-regulating kinase homologue MoArk1 plays a pleiotropic function in Magnaporthe oryzae. Mol Plant Pathol 2013, 14, 470-482.

Point 4: “In order to further verify the regulatory effect of citral on the chitin synthesis pathway in M. oryzae, we selected eight DEGs (six upregulated genes and two downregulated genes)…”What is the reason of this selection?

Response 4: Thanks very much for your attention to our paper! The co-expressed genes of amino sugar and nucleotide sugar metabolic pathways in M.oryzae were screened under the action of citral at the concentration of 50 and 100 ug/mL for 24 h. Through Ven diagram analysis and GO analysis, 8 genes affecting the cell wall of M.oryzae were finally selected from the amino sugar and nucleotide sugar metabolic pathways(Supplementary Fig. S3).

References are as follows:

Park, C.H.; Kim, S.; Park, J.Y.; Ahn, I.P.; Jwa, N.S.; Im, K.H.; et al. Molecular characterization of a pathogenesis-related protein 8 gene encoding a class III chitinase in rice. Mol Cells 2004, 17, 144-150.

Davies, J.S.; Coombes, D.; Horne, C.R.; Pearce, F.G.; Friemann, R.; et al. Functional and solution structure studies of amino sugar deacetylase and deaminase enzymes from Staphylococcus aureus. FEBS Let 2019, 593, 52-66.

Groeve, M.R.M.D.; Depreitere, V.; Desmet, T.; Soetaert, W. Enzymatic production of alpha-D-galactose 1-phosphate by lactose phosphorolysis. Biotechnol Lett 2009, 31, 1873-1877.

Point 5: “The results show that the antifungal mechanism of citral against M. oryzae occurs via regulating the expression of key genes involved in the chitin and UDP glucose synthesis pathways.” Are there mutants known from literature which support those results?

Response 5: Author would like to thank constructive comments of reviewer. our previous research results showed that citral had a strong antibacterial activity against M. oryzae. The results of enzyme activity and proteomics determination showed that citral may increase the chitinase activity of M. oryzae, destroy the structure of cell wall and inhibit the growth of M. oryzae. Therefore, our study via that the RNA-Seq Analysis and real-time fluorescence quantitative verification that the citral could affect chitin synthesis and UDP glucose synthesis in the amino sugar and nucleotide metabolic pathways of M. oryzae. RT-qPCR analysis the chitin present in M. oryzae might be degraded to chitosan, chitobiose, N-acetyl-D-glucosamine, and β-D-fructose-6-phosphate following treatment with citral. Chitin degradation was indicated by damaged cell-wall integrity. While the UDP glucose synthesis pathway was involved in glycolysis and gluconeogenesis, providing precursors for the synthesis of polysaccharides. Galactose-1-phosphate uridylyltransferase involved in the regulation of UDP-α-D-galactose and α-D-galactose-1-phosphate was downregulated, which would result in the inhibition of UDP glucose (UDP-Glc) synthesis, a reduction in cell-wall glucan content, and the destruction of cell-wall integrity. At present, our research group is constructing silencing vectors for the target genes were screened in this study, and observing the genotype expression and other related work in the later stage.

References are as follows:

①    Li, R.Y.; Wu, X.M.; Yin, X.H.; Liang, J.N.; LI, M.The Natural Product Citral Can Cause Significant Damage to the Hyphal Cell Walls of Magnaporthe grisea. Molecules 2014, 19, 10279-10290.

②    Li, R.Y.; Wu, X.M.; Yin, X.H.; Long, Y.H.; Li, M. Naturally produced citral can significantly inhibit normal physiology and induce cytotoxicity on Magnaporthe grisea. Pestic Biochem Physiol 2015, 118, 19-25.

③    Zhao QJ.; Ding, Y.; Song, X.C.; Liu, S.J.; Li, M.; LI, R.Y.; et al. Proteomic analysis reveals that naturally produced citral can significantly disturb physiological and metabolic processes in the rice blast fungus Magnaporthe oryzae. Pestic Biochem Physiol 2021, 175, 104835.

Point 6: on the choice of colors in Fig. 7, which makes the model's descriptions hard to read.

Response 6: Thanks very much for your attention to our paper! I had amended it. The following was a detailed explanation:

Figure 7. Model diagram illustrating the mechanism of citral against the M. oryzae cell wall.

Reviewer 4 Report

In this manuscript authors check The Antifungal Effects of Citral on Magnaporthe oryzae Occur via Modulation of Chitin Content as Revealed by RNA-Seq Analysis. Authors screen and annotate genes that are differentially expressed (also known as DEGs) in M. oryzae after treatment with citral using RNA sequencing (RNA-seq). A total of 21502 differentially DEGs were identified, including 15083 downregulated and 6419 upregulated genes. In particular, DEGs in M. oryzae were significantly affected after treatment with citral, with 2301 DEGs (1392 downregulated and 909 upregulated genes) common to all concentrations and time points of citral treatment, and 649 DEGs specifically for 100 μg/mL citral treatment for 24 h. Most identified DEGs participate in the biosynthesis of mycelium development, conidium formation, and transmembrane transport process. KEGG pathway analysis showed that DEGs were enriched in the amino sugar and nucleotide sugar metabolism pathways of M. oryzae, which are related to chitin formation. Authors present a model diagram illustrating the mechanism of citral in acting against the M. oryzae cell wall. The differential main expression of genes (MGG_10333, MGG_01336, MGG_00625) in the chitin formation pathway of M. oryzae following treatment with citral could lead to a reduction in chitin content and cell wall damage through the inhibition of chitin synthesis. Furthermore, the differential expression of genes (MGG_04495, and MGG_05098) in the uridine diphosphate (UDP) glucose formation pathway of M. oryzae after treatment with citral could lead to a disruption in cell-wall integrity, through inhibition of UDP glucose synthesis and a reduction in glucan content of the cell wall, thus inhibiting mycelial growth. The manuscript looks refined now and can be accepted for publication after minor grammar check.

Author Response

November 10,2021

Dear Reviewer

Thanks very much for your attention and consideration to our manuscript! In revised manuscript, we have taken into consideration every comment/question raised by the reviewers, and made corresponding changes. Our point-by-point responses to the reviewers' comments are summarized in an itemized list.

We hope that this revision can be considered for publication by Journal of Fungi. Please feel free to contact me at +86 15185148063 by phone, or [email protected].

Thank you very much for your attention and consideration.

Best wishes,

Rong-Yu Li

Institute of Crop Protection, Guizhou University

Jiaxiunan Road 515, Huaxi District, Guiyang, 550025,Guizhou, China

Phone: +86-15185148063

Response to Reviewer 4 Comments

Point 1: The manuscript looks refined now and can be accepted for publication after minor grammar check.

Response 1:Thanks very much for your attention to our paper! I have checked the grammar of my article. For details, please refer to the uploaded original manuscript.

Round 2

Reviewer 1 Report

I think that the manuscript now has been improved and can be accepted for publishing.
The authors replied perfectly on the comments and the paper contains the fungicidal effect of Citral on Magnaporthe oryzae and include also some molecular analysis interesting.
I accept the article in its current form.

Reviewer 2 Report

Thanks for providing the revised version of the manuscript.

the current version looks much better with much improvement. However, I still have two minor comments before moving to the next step of accepting this manuscript for publication.

1-  The title MUST be changed. The authors claimed that “The antifungal effects of citral on M. oryzae occur via modulation of chitin content”, however, I did not find any quantitative, or even qualitative work, for chitin content throughout the manuscript. All that I see is only some differentially expressed genes (DEGs), some of them are related to chitin synthesis. please remove the "chitin content" from the title or replace it with something more related to your findings.

2- in Figures 2, 3, and 5, add the panel lettering (A, B, C, …..etc.) to the upper left corner of each panel.

Author Response

November 10,2021

Dear Reviewer

Thanks very much for your attention and consideration to our manuscript! In revised manuscript, we have taken into consideration every comment/question raised by the reviewers, and made corresponding changes. Our point-by-point responses to the reviewers' comments are summarized in an itemized list.

We hope that this revision can be considered for publication by Journal of Fungi. Please feel free to contact me at +86 15185148063 by phone, or [email protected].

Thank you very much for your attention and consideration.

Best wishes,

Rong-Yu Li

Institute of Crop Protection, Guizhou University

Jiaxiunan Road 515, Huaxi District, Guiyang, 550025,Guizhou, China

Phone: +86-15185148063

Point 1: The title MUST be changed. The authors claimed that “The antifungal effects of citral on M. oryzae occur via modulation of chitin content”, however, I did not find any quantitative, or even qualitative work, for chitin content throughout the manuscript. All that I see is only some differentially expressed genes (DEGs), some of them are related to chitin synthesis. please remove the "chitin content" from the title or replace it with something more related to your findings.

Response 1:Thanks very much for your attention to our paper! I have marked text in red. The following was a detailed explanation:

The Antifungal Effects of Citral on Magnaporthe oryzae Occur via Modulation of Key enzymes in cell wall as Revealed by RNA-Seq Analysis

Point 2: in Figures 2, 3, and 5, add the panel lettering (A, B, C, …..etc.) to the upper left corner of each panel.

Response 2: Author would like to thank constructive comments of reviewer. I had marked the text in red. The following was a detailed explanation:

Figure 2. Volcanic map of DEGs in M. oryzae after treatment with citral at different concentrations(A-F). Red represents upregulated significantly differentially expressed genes, blue represents downregulated significantly differentially expressed genes, and gray dots represent non significantly differentially expressed genes.

Figure 3. Volcanic map analysis of DEGs in M. oryzae after treatment with citral for 24 h(A,B). Red represents upregulated significantly differentially expressed genes, blue represents downregulated significantly differentially expressed genes, and gray dots represent non significantly differentially expressed genes.

Figure 4. Summary of the annotations of DEGs in M. oryzae. (A) GO classifications. (B) Scatter plot of KEGG enrichment of DEGs. (C) Enrichment of GO genes in M. oryzae after treatment with citral (100 μg/mL) for 24 h.

Figure 5. RT-qPCR verification of key genes of resistance to citral in M. oryzae. (A,B) The expression trend of genes as characterized by qPCR is consistent with that of RNA-seq analysis after 9, 12, and 24 h treatment with citral (100 μg/mL). (C,D) The expression trend of genes as characterized by qPCR is consistent with that of RNA-seq analysis after treatment with 0, 50, and 100 μg/mL citral for 24 h. All values are based on three technical repeats and presented as means±SD. Different characters indicate a statistically significant difference at p< 0.05 by t-test.

Reviewer 3 Report

Dear authors,

thank you for the detailed point-by-point response and for taking the recommendations into account.

Round 3

Reviewer 2 Report

Thanks for addressing my previous comments and suggestions.

This manuscript is a resubmission of an earlier submission. The following is a list of the peer review reports and author responses from that submission.